# Surface-Initiated Polymerization with an Initiator Gradient: A Monte Carlo Simulation

**DOI:** 10.3390/polym16091203

**Published:** 2024-04-25

**Authors:** Zhining Huang, Caixia Gu, Jiahao Li, Peng Xiang, Yanda Liao, Bang-Ping Jiang, Shichen Ji, Xing-Can Shen

**Affiliations:** 1State Key Laboratory for Chemistry and Molecular Engineering of Medicinal Resources, Key Laboratory for Chemistry and Molecular Engineering of Medicinal Resources, Ministry of Education of China, Collaborative Innovation Center for Guangxi Ethnic Medicine, School of Chemistry and Pharmaceutical Sciences, Guangxi Normal University, Guilin 541004, China; huang_z_n@stu.gxnu.edu.cn (Z.H.); jiangbangping@mailbox.gxnu.edu.cn (B.-P.J.); 2School of Computer Science and Engineering & School of Software, Guangxi Normal University, Guilin 541004, China; lyd@gxnu.edu.cn; 3State Key Laboratory of Molecular Engineering of Polymers, Fudan University, Shanghai 200433, China

**Keywords:** gradient brush, surface-initiated polymerization, stochastic reaction model, Monte Carlo simulation

## Abstract

Due to the difficulty of accurately characterizing properties such as the molecular weight (*M*_n_) and grafting density (*σ*) of gradient brushes (GBs), these properties are traditionally assumed to be uniform in space to simplify analysis. Applying a stochastic reaction model (SRM) developed for heterogeneous polymerizations, we explored surface-initiated polymerizations (SIPs) with initiator gradients in lattice Monte Carlo simulations to examine this assumption. An initial exploration of SIPs with ‘homogeneously’ distributed initiators revealed that increasing *σ* slows down the polymerization process, resulting in polymers with lower molecular weight and larger dispersity (*Đ*) for a given reaction time. In SIPs with an initiator gradient, we observed that the properties of the polymers are position-dependent, with lower *M*_n_ and larger *Đ* in regions of higher *σ*, indicating the non-uniform properties of polymers in GBs. The results reveal a significant deviation in the scaling behavior of brush height with *σ* compared to experimental data and theoretical predictions, and this deviation is attributed to the non-uniform *M*_n_ and *Đ*.

## 1. Introduction

Gradient brushes (GBs) are polymer brushes wherein properties, such as molecular weight, grafting density, or chemical composition, gradually vary in one or more directions along the substrate. GBs are powerful tools for high-throughput and low-cost investigations in the areas of physics, chemistry, biomaterials science, and biology [1,2,3,4,5,6,7,8]. In a single sample, a given surface parameter across a wide range can be systematically explored, avoiding the need for lengthy repetitive procedures and enhancing the efficiency of research and development [8]. Additionally, GBs are widely used to study interfacial phenomena like the directional transport of liquids, cell adhesion, and migration [9,10]. 

Surface-initiated polymerization (SIP) is a promising approach to synthesizing GBs with higher grafting density. There are two major forms of SIP classified by initiator distribution [3,11]: one with a homogeneous distribution of initiators and another with a gradient distribution. In the former case, GBs are obtained by controlling the spatial polymerization time, for example, using a movable mask or reaction solution, or by adjusting the spatial polymerization rate through methods such as varying the intensity of transmitted light with a filter in photopolymerization [12,13,14,15,16]. In the latter case, initiators with a gradient density are firstly anchored to the surface using methods such as the silane diffusion method, nanolithography methods, or methods involving gradients of temperature or electrochemical potential [15,16,17,18,19,20,21]. Subsequently, SIP is carried out to yield GBs. Generally, the former is a simpler and more feasible method, while the latter is suitable for small-sized patterns and arbitrary structures [3].

Despite significant progress in the preparation of GBs, characterizing crucial properties such as grafting density (*σ*), molecular weight (*M*_n_), and molecular weight distribution remains a challenging task [13,22,23]. The characterization of *σ* and *M*_n_ is interrelated since
(1)$$\sigma =h\rho N_{A}/M_{n},$$
where *h* is the height of a brush in the dry state, *ρ* is the density of a polymer, and *N*_A_ is Avogadro’s constant. Gel permeation chromatography (GPC) is the most common method for directly determining the molecular weight of grafted polymers, involving the degrafting of polymer chains from a substrate. However, the GPC method requires a sufficient amount of a sample, posing a challenge for SIP, especially regarding polymerization on a flat substrate [2]. On the other hand, the accuracy of indirect measurement, achieved by incorporating sacrificial initiators for simultaneous bulk- and surface-initiated polymerizations and characterizing the resulting polymers in solution, has been a topic of debate [22,23]. Notably, neither direct nor indirect GPC methods offer insights into the spatial distribution of these properties in GBs.

The lack of information on molecular weight and grafting density has significantly hindered efforts toward comprehensively understanding gradient polymer brushes and applying them. An early study examined the scaling behavior between brush height and grafting density for a gradient polymer brush [17]. In this study, a polyacrylamide (PAAm) brush with a grafting density gradient was obtained via atom-transfer radical polymerization (ATRP) with an initiator gradient generated via the silane diffusion method [17,24]. Due to the absence of information on molecular weight, two assumptions were made to determine the grafting density in space: (1) the molecular weight of polymers along the substrate in GBs is uniform, and (2) there is similarity in the molecular weight between polymers in the GBs and those obtained in solution polymerization under the same conditions. Upon determining the dry brush height using variable-angle spectroscopic ellipsometry, the spatial distribution of grafting density was obtained using Equation (1). Subsequently, we examined the scaling relationship between the wet height of the brush and grafting density, revealing only a slight deviation from the theoretical prediction [17].

Although researchers have recognized the limitations of the uniform molecular weight assumption [17], it remains prevalent in experimental studies due to its simplicity. This raises a question: what is the significance of the effect of this assumption?

To answer this question, the polymerization mechanism should be examined, as it strongly influences properties such as molecular weight and dispersity. Computer simulations have played an important role in revealing the mechanisms of SIPs. A pioneering study was performed by Genzer using a Monte Carlo (MC) simulation [25], and it inspired studies using different simulation methods [26,27,28,29,30,31,32,33,34,35]. Typically, SIPs result in polymers with larger dispersity and smaller molecular weight compared to those generated via bulk-initiated polymerizations (BIPs). This trend holds even in simultaneous bulk- and surface-initiated polymerization [26,28,35]. Moreover, grafting density is a key parameter in SIP, impacting both the kinetics of the reaction and the properties of the polymers, such as molecular weight and dispersity. The main reason is that SIP is a heterogeneous polymerization, as the homogeneous distribution of free monomers is altered by the newly formed polymer brush [34], while BIP is a homogeneous polymerization. 

According to the existing simulations of SIPs, it is natural to expect that the molecular weight should be non-uniform in an SIP with an initiator gradient. However, the significance of this difference in molecular weight and its potential impact on the scaling behavior observed in Ref. [17] remain uncertain because no simulations, to the best of our knowledge, have directly addressed this issue.

To address this gap, we conducted a lattice Monte Carlo simulation to examine SIP with initiator gradients (referred to as gradient polymerization in the remainder of this paper), using a stochastic reaction model (SRM) developed for heterogeneous polymerizations [34,35,36]. We explored two systems: one with a series of homogeneous SIPs with varying grafting densities, wherein the initiators were homogeneously distributed, and the other with SIPs with an initiator gradient. In both systems, the properties of the polymers are significantly affected by grafting density. Notably, the scaling relationship between the brush height and grafting density in GBs, as obtained in the simulation, diverges from the experimental results, highlighting the need for a more in-depth exploration of gradient polymerization. The lattice MC model and SRM algorithms are introduced in Section 2, while the results of living polymerizations with homogeneous and gradient initiators are shown in Section 3. Brief conclusions are provided in Section 4.

## 2. Models and Simulation Methods

### 2.1. Lattice Monte Carlo Simulation

In this study, we employed the Larson-type bond fluctuation model [37,38], which was previously utilized in our investigations of SIP and the flow behavior of polymer brushes [34,35,36,39]. Briefly, the simulation was carried out in a simple cubic lattice with a volume, *V*, equal to *L_x_* × *L_y_* × *L_z_*. Each lattice site can be occupied by a monomer or initiator only once, and the bond length was set to 1 or √2. During relaxation, a monomer is randomly selected to be exchanged with one of its 18 nearest or next-nearest neighbor sites. This exchange will be accepted under the conditions that the neighbor site is vacant and that the exchange would not break the chain and cause bond intersection (possible bond intersections are shown in Appendix A). The excluded volume effect and entanglement were well considered in this model. The simulation time was measured in units of Monte Carlo steps (MCs), defined as all monomers attempting to move once, on average. 

Two impenetrable walls were set in the *y* = 1 and *L_y_* planes, respectively, while periodic boundary conditions were applied in both the *x* and *z* directions. In this simulation, the *x* direction represents the initiator gradient direction, the *y* direction indicates the chain growth direction, and the *z* direction corresponds to the equivalent direction. In the beginning, all the free monomers were randomly distributed in the system. The immobilized initiators were randomly positioned on the *y* = 1 plane during the investigation of homogeneous SIP. It should be noted that the term ‘homogeneous’ does not imply a perfectly ‘regular’ distribution of initiators, as shown in Appendix A. The properties of these two systems show subtle differences [30]. Instead, “homogeneous” is used in comparison to the gradient polymerization. 

While studying gradient polymerization, the *y* = 1 plane was divided into multiple stripes in the *x* direction (initiator gradient direction), as illustrated in Figure 1. Each stripe has a width *w*, resulting in a total number of *L_x_*/*w* stripes. In the left part of the simulation box, the grafting density of the leftmost stripe is *σ*_min_ and gradually increases with the value of Δ*σ* (the difference in density between successive stripes) with an increasing number of stripe locations until it reaches the maximum grafting density *σ*_max_. The grafting density in the right part mirrored that of the left part, i.e., the grafting density decreased from the middle to the rightmost stripe as the location of stripes shifted forward further. Within each stripe, the number of initiators can be calculated as *σ*(*x*) × *w* × *L_z_*, with *σ*(*x*) denoting the grafting density of a stripe, and these initiators exhibited a random distribution within the stripes. 

### 2.2. Implementation of Polymerization

In this study, a living polymerization was considered, which occurred at intervals of every *τ* MCs during the relaxation process. Here, *τ* is defined as the characteristic delay time, or reaction interval time [27,34,35,36,40]. By decreasing or increasing the value of *τ*, the reaction can be adjusted to make it diffusion-limited or reaction-limited. 

The stochastic reaction model (SRM) proposed by our group was applied to model the polymerization [34,35,36]. Firstly, an initiator or active center was randomly selected. Then, the number of free monomers *m* within the radius of √2 was determined. The initiator or active center tries to react with a random monomer among these *m* free monomers with a given probability *P*_r_, which is determined by the local number of free monomers and calculated as *mP*_0_ (where *P*_0_ is a constant representing the elementary reaction probability between one active center and one free monomer). If the reaction is accepted, the free monomer transforms into an active center for future reactions. Since *P*_r_ is determined by the local reaction environment, each active center reacts with its own probability. Thus, our SRM model fully accounts for the heterogeneous reaction microenvironment, which is the inherent character of SIP [34,35,36]. As demonstrated by living bulk-initiated polymerization, the polymerization kinetics obtained by this SRM were found to be very consistent with the theoretical predictions [34,36].

## 3. Results and Discussion

### 3.1. Homogeneous Surface-Initiated Polymerization

We first investigated a series of homogeneous SIPs with varying grafting densities and compared the properties at the same polymerization times. The parameters were fixed and set as follows: the dimensions of the simulation box were *L_x_* × *L_y_* × *L_z_* = 50 × 77 × 50, the initial concentration of free monomers was [*M*]_0_ = 0.4 monomers per lattice, the reaction interval time was *τ* = 10 MCs, the simulation time was 2 × 10^6^ MCs, and the elementary reaction probability was *P*_0_ = 0.001. The results were averaged over 60 independent runs.

Figure 1a shows the number-average molecular weight *M*_n_ during polymerization with a given grafting density *σ.* It is evident that *σ* significantly influences the polymerization kinetics, with *M*_n_ increasing more rapidly at lower *σ* compared to higher values. Figure 1b shows *M*_n_ as a function of *σ* with a given polymerization time *t*. When *σ* is very low, *M*_n_ is nearly constant. However, when *σ* is high, *M*_n_ exhibits a monotonic decrease with an increasing *σ*. The decrease in *M*_n_ with increasing *σ* is related to the polymerization time. For example, when *t* = 400,000 MCs, the values of *M*_n_ are 21.7 and 13.5 at *σ* = 0.1 and 0.4, respectively. The latter (*M*_n_ = 13.5) is only about 62% of the former, and by *t* = 800,000 MCs, this ratio further decreases to 54%. Besides *M*_n_, the dispersity (*Đ* = *M*_w_/*M*_n_) and molecular weight distribution (MWD) are also influenced by *σ*. The dispersity increases with increasing *σ* (Figure 1c), and MWD becomes broader and more asymmetric (Figure 1d). 

The results suggest that in SIP, systems with different values of *σ* exhibit variations in polymer properties at the same polymerization time, preliminarily indicating that the molecular weight in GBs might be non-uniform due to the initiator gradient. It should be pointed out that molecular weight is independent of the concentration of initiators in living BIPs with low monomer conversion, and this can be proved as follows. The monomer conversion *C* of BIP can be written as
(2)$$C=1-\frac{{\left[M\right]}_{t}}{{\left[M\right]}_{0}}=1-\exp(-(m_{\max}-1){\left[I\right]}_{0}P_{0}t/\tau ),$$
where [*M*]*_t_* is the concentration of free monomers at time *t*, [*I*]_0_ is the concentration of the initiator, and *m*_max_ is the maximum number of free monomers around an active center, equal to 18 in this simulation. At low conversion, *M*_n_ linearly increases with *t* as
(3)$$M_{n}=\frac{{\left[M\right]}_{0}C}{{\left[I\right]}_{0}}\approx {\left[M\right]}_{0}(m_{\max}-1)P_{0}t/\tau .$$

Thus, in BIP, molecular weight is only determined by reaction time, and it is independent of the number of initiators. 

Why does this assumption hold for BIP but not SIP? The reason is that Equation (1) was deduced from a homogeneous polymerization system, such as BIP. However, SIP is a heterogeneous system, especially when *σ* is high. When *σ* is low, the active centers are far apart and react with free monomers like isolated active centers, resembling a homogeneous polymerization system. Conversely, when *σ* is large, brush-like polymers are obtained, and the system is no longer homogeneous as free monomers are expelled by the nascent polymers from the surface. The active centers near the surface react in an environment with a lower concentration of free monomers compared with those far from the surface. Such a heterogeneous reaction environment is the key feature of SIP, and the heterogeneity of the reaction environment increases with *σ* [34,35]. 

### 3.2. Surface-Initiated Polymerization with Initiator Gradient

We further investigated the gradient polymerization with a simulation box for which *L_x_* × *L_y_* × *L_z_* = 288 × 72 × 100. The initial concentration of free monomers was [*M*]_0_ = 0.4 monomers per lattice. As shown in Figure 1, the grafting plane was divided into stripes with a width *w* = 4. In this study, the maximum grafting density of a stripe is *σ*_max_ = 0.42, and the minimum is *σ*_min_ = 0.07. We did not explore lower grafting densities as our primary interest lay in the scaling behavior of polymer brushes within regions with high grafting density. The difference in grafting density between the adjacent stripes is Δ*σ* = 0.01. The corresponding steepness of the gradient is δ = Δ*σ*/*w* = 0.0025. As discussed later, such a steepness is low enough to examine the gradient polymerization process. Polymerization stops when the number-average molecular weight of the brush reaches 50. Such a molecular weight is large enough to ensure the system stays in the brush region; meanwhile, it can avoid the situation wherein some very long chains might approach the *y* = *L_y_* plane.

The density contour map (Figure 2a) clearly confirms the formation of a gradient brush. In the vertical direction, the polymer density decreases with an increasing distance from the surface. Horizontally, there is a gradient increase in the density from the low-grafting region (*x* = 1) to the high-grafting region (*x* = 144), which then decreases upon further shifting the *x* position forward. Meanwhile, the density contour map of free monomers exhibits the opposite trend (Figure 2b). 

The molecular weights of the polymers at each stripe were examined (Figure 3a), with the results clearly proving that there is a non-uniform molecular weight in GBs. *M*_n_ decreases from the low-grafting regions (outsides) to the high-grafting regions (middle). Although the overall *M*_n_ of the system is 50, it is only 42.4 in the middle, contrasting with the higher value of 65.5 on the outside areas. Such a notable difference in *M*_n_ should not be overlooked. We further compared the molecular weight and dispersity in gradient polymerization with those in SIP at the same reaction time (Figure 3b). The variation in *M*_n_ between different grafting density positions in gradient polymerization is smaller than that in SIP. This might be attributed to the competition among different polymerization regions in gradient polymerization. Higher-grafting-density regions tend to consume more monomers during the reaction. The dispersity of polymers in gradient polymerization increases with grafting density, which is similar to the trend observed in SIP (Figure 3b).

In experiments, the height of a brush in the dry state is widely used to estimate grafting density in accordance with Equation (1). In this study, we have supposed that the brush shown in Figure 2a vertically collapses onto the surface; thus, the dry height at position *x*, denoted as *h*(*x*), can be calculated as follows:(4)$$h(x)=\sum_{y=1}^{L_{y}}\rho (x,y).$$

Figure 4a suggests that a gradient brush is obtained but that the dry height *h*(*x*) does not linearly increase with the grafting density *σ*(*x*), as depicted by Equation (1). This deviation from a linear relationship is evidently caused by the variations in *M*_n_ at different positions, as shown in Figure 3a. After normalizing the height with respect to the corresponding molecular weight, a linear relationship between *h*(*x*) and *σ*(*x*) can be restored. 

The height in solution *H* is a key property of a polymer brush and is calculated as follows:(5)$$H(x)=\sum_{y=1}^{L_{y}}y\rho (x,y)/\sum_{y=1}^{L_{y}}\rho (x,y)$$

Figure 4b suggests that *H* increased from the outside areas to the middle, indicating the formation of a gradient brush. We are more interested in the relationship between *H*(*x*) and *σ*(*x*). In the low-grafting-density region, *H* increases only slightly with *σ* (Figure 4c). Subsequently, a scaling relationship between them with a scaling exponent of 0.15 can be observed. In the even-higher-grafting-density region, the increase in thickness slows down again. We speculate that the absence of pronounced scaling behavior on both sides is due to the unidirectional extension of the chains. In the region with the highest grafting density, the chains primarily extend towards the low-density region due to the significant compression between chains. The opposite behavior is observed in the low-density region. This explanation is supported by Appendix A. In the high (low)-grafting-density region, the number of monomers consumed by corresponding initiators during polymerization is greater (less) than the actual number of monomers in that region. 

If we focus on the scaling behavior in the middle region, the obtained scaling exponent (0.15) is much smaller than both the theoretical value (1/3) [41,42] and the experimental value (0.37–0.4) obtained in Ref. [17]. The theoretical scaling exponent has been extensively validated through simulations [39,43,44,45,46], although some simulations suggest that it may not be a constant but instead slightly vary with the grafting density [47]. This raises the following question: how can we interpret such a smaller scaling exponent obtained in this simulation?

First, the theoretical scaling exponent of one-third was validated for brushes of monodisperse polymers in good solvents. As shown in Figure 3a, in gradient polymerization, *M*_n_ decreases with an increasing grafting density in the *x* direction, and the difference in *M*_n_ is significant. Moreover, the polymers at each stripe are polydisperse, and the dispersity increases with the increase in grafting density. Studies have shown that the dispersity of polymers also influences the height of polydisperse polymer brushes [34,48,49]. Thus, in a gradient polymerization system, if we only examine the relationship between height and grafting density, the scaling exponent does not need to adhere to the one-third scaling behavior. We are particularly interested in the experimental value [17], as the non-uniform molecular weight and the dispersity should also be present in the experiment as in this simulation. The fact that the experimental value is close to the theoretical prediction might be related to the method used to calculate the grafting density or instead be a coincidental approximation. 

The observed small exponent might relate to the steepness of the initiator gradient. In GBs, polymers experience unbalanced lateral compression in the gradient direction, causing chains in high-grafting-density regions to extend towards regions with lower grafting density. In the experiment, the steepness is negligible as the lateral size of the substrate is 10^6^ times larger than the height of the polymer brush [17], while in simulations, the role of steepness should be considered due to the finite simulation size. We can expect that the larger the steepness, the smaller the exponent. 

To address this, we studied GBs with different levels of steepness (Figure 4d). It should be pointed out that these GBs consist of polymers with a fixed value of *M*_n_ = 50, avoiding the influence of non-uniform molecular weight and dispersity observed in gradient polymerization. Figure 4d shows the relationship between brush height and grafting density. When the steepness decreases from 0.005 to 0.0025, the curve exhibits a steeper incline, indicating the influence of steepness. A further reduction in steepness to 0.00125 causes only a minor change in the curve. In this case, the scaling exponent in the middle region is about 0.3, close to the theoretical value. Additionally, we varied the stripe width from *w* = 4 to *w* = 1 while fixing the steepness (Figure 4d), revealing almost identical results. Thus, we believe that the applied steepness is small enough, and this factor is not the main cause of the observed small scaling exponent.

Although the steepness cannot be infinitely small in simulations, we can examine a series of homogeneous SIPs with different initiator densities, the reaction time of which is the same as that of gradient polymerization. This special case allows us to approximate the behavior of gradient brushes with zero steepness. Figure 4c demonstrates that the heights of brushes obtained through SIP display a certain scaling relationship, with a scaling exponent (0.17) slightly larger than that of gradient polymerization (0.15). Thus, we can conclude that the steepness is not the primary cause of the observed small scaling exponent in our simulation.

## 4. Conclusions

The application of gradient brushes requires critical information regarding properties such as molecular weight and grafting density, which are difficult to characterize experimentally. The assumption that polymers in gradient brushes have uniform properties was commonly adopted in previous experiments, and its validity was questioned but not directly examined. In this study, we employed a stochastic reaction model to investigate surface-initiated polymerization with initiator gradients and analyzed the properties of the resulting gradient brushes.

We first examined surface-initiated polymerizations with homogeneously distributed initiators. The results indicated that, at a given reaction time, polymers with lower molecular weight and higher dispersity were obtained when there was increasing grafting density. This trend can be attributed to the heterogeneous reaction environment inherent in SIP. Similarly, in SIPs with an initiator gradient, the corresponding polymers exhibited position-dependent properties, with lower molecular weights and higher dispersity at positions with higher grafting density. The difference in molecular weight in gradient brushes, reaching up to 154% (65.5/42.4) in this study, strongly supports the notion that the properties of polymers in gradient brushes are non-uniform.

Subsequent investigation into the height of gradient brushes in solution revealed a small scaling exponent (0.15) in scaling behavior with respect to grafting density, notably deviating from the expected scaling exponent of 1/3. We attributed this discrepancy to the variations in molecular weight and dispersity across space while also excluding the influence of the steepness of initiator gradient. It is noteworthy that a 1/3 scaling exponent is conventionally applied to monodispersed polymer brushes.

We are intrigued by the proximity of the experimental scaling exponent to the theoretical value since the non-uniform properties of polymers should also be present in an experiment. However, we failed to find any other experimental studies of scaling behavior with respect to gradient brushes, and we are uncertain whether this behavior is just a coincidence or if there are underlying mechanisms. It is important to recognize the differences between the simulation and experiments. In the simulation, a living polymerization was modeled, and all initiators reacted, while in the experiment, ATRP was applied [17], and the initiator efficiency was typically low [23].

In summary, this study provides direct evidence of the significant non-uniform properties of polymers in surface-initiated polymerizations with initiator gradients. We hope experimental studies are conducted in the future to better clarify the experimental results in Ref. [17]. Additionally, it would be interesting to investigate surface-initiated polymerizations with other gradients, such as reaction time [13]. 

## Data Availability

Data are contained within the article and Appendix A.

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
