# Peer review of "Surface-Initiated Polymerization with an Initiator Gradient: A Monte Carlo Simulation"

_polymers, 2024, doi:10.3390/polym16091203_

Round 1
Reviewer 1 Report
Comments and Suggestions for Authors
GThe paper is mainly an in-silico modeling of surface-initiated polymerization (SIP) of gradient polymer brushes. The authors utilize their lattice-based Monte Carlo framework involving stochastic reaction modeling (SRM) to simulate the SIP. The authors demonstrate that calculating the height in a more precise /granular manner since it is a gradient brush system (by not considering the assumption of uniform molecular weight distribution), definitely brings in a new characterization feature of the gradient brush system. However, after reading the manuscript, I have several concerns with the manuscript (see below), and thus do not recommend the manuscript for publication in its present form.
The major concern is related to the way the paper is written.
Line 71 - 75: The authors themselves clearly state that the two assumptions were described in ref [17],[24]. Further, there was close agreement between theory and experiment.
However, the authors go on to state: "According to the simulation results, it becomes apparent that the assumptions made to calculate grafting density in Genzer's study are not appropriate, potentially leading to an inaccurate scaling relationship analysis. "
I understand that the assumption of uniform molecular weight does not depict the real scenario. However, I request the authors to tone down their claims such as “invalid”, and "inaccurate". The earlier authors were aware of the fact and thus they stated those points as assumptions. Thus, it is my earnest request to the authors to show academic integrity and rewrite the paragraphs where the authors have criticized previous work. It does not read well and seems like a “personal attack” on one of the cited authors. The authors should demonstrate the shortcomings of earlier research and the limitations but those criticisms should not sound like a personal attack. Thus, I request the authors to maintain academic politeness and rewrite the paragraphs and sentences highlighting the shortcomings without getting too personal with names.
Thus, I would request the authors to not make strong remarks and modify such sentences by stating that they are providing an improved characterization method or they believe that their characterization method can bring about significant improvement in the understanding of gradient brushes. The authors of the current manuscript however have not made any mean-field theoretical justification for their scaling results or related their results to some experimental findings (I understand that the purpose of this paper is to request future experiments). However, they fail to explain how characterizing the H instead of h can provide significant advancement for the experimentalists.
In line 301 - 305 “The applications of gradient brushes require critical information regarding properties such as molecular weight and grafting density, which are challenging to characterize experimentally. To overcome these limitations, we employed a stochastic reaction model to investigate surface-initiated polymerization with initiator gradients and analyzed the properties of the resulting gradient brushes."
It is not clear how this modeling has helped achieve overcome this limitation. The authors have not shown any correlation or theoretical formulation to corroborate that their modeling can provide insights into the experiments, and help overcome the limitations of the experiments.
Further, the authors use words like: “Consequently, the widely adopted assumption of uniform molecular weight for estimating grafting density in experiments is invalidated.”
“we believe that the scaling exponent obtained in the experiment is merely an approximate coincidence with the theoretical value.” The use of words such as “believe”, and “approximate coincidence”, without providing a proper quantitative explanation is insufficient to make such claims of invalidity.
Also, please find below further queries based on this work. Thus, in conclusion, I cannot recommend the manuscript for publication in its present form, and request the authors to carry out major revisions to the manuscript.
Queries:
- Line 18: "SIP's" use of abbreviation without the declaration of the full form. Please provide the full form.
- The authors are requested to change the title of their manuscript. My issue is with the second part of the title. The current article is not a review article but an article of original research. The use of the sentence: "A System Calling for In-depth Study" sounds like a cry for an in-depth study and thus, fails to justify why this paper is calling for an in-depth study and why the authors themselves have not carried them out.
- Further, the last sentence of the abstract (lines 24 -25): "This highlights the need for in-depth studies in the future to a better understanding SIP with initiator gradient." It is not clear why the authors themselves have not carried them out. If it is a call focused on experiments, then the authors are requested to revise the sentence.
- In the conclusions section, Lines 325 - 326 "In summary, this study reveals the complexity of surface-initiated polymerizations with initiator gradients, emphasizing the need for more comprehensive investigations to deepen our understanding". It is a vague statement which does not provide any detail. Please add more details including how this can be achieved. Otherwise, the paper seems to fail to provide any additional guidelines for improving the study of gradient brushes, except for their granular calculations of brush height vs grafting density. How can that be achieved experimentally, and how this information can be utilized to improve the applicability of the gradient brushes has not been conveyed in the paper except for superlative claims without any clear citation or quantitative reasoning.
- In the abstract, in the first sentence in line 14: The authors state that “widespread applications of gradient brushes (GBs),” while in line 65, it is stated "hindered …. the application of gradient polymer brushes". Please resolve the contradictory statements or rewrite them to bring out the meaning.
- Line 14: "Despite the widespread applications of gradient brushes (GBs), accurately characterizing properties like molecular weight (Mn) and grafting density ($\sigma$) in space remains a challenging task in experiments." In line 15, the authors just state their in-silico model details. The way the first sentence is framed, it seems in the present manuscript, the authors are going to provide ways that will help overcome the stated 'challenging tasks' in experiments. But, the paper has a completely different conclusion. Thus, I request the author to rewrite the abstract that specifically meets the details provided in the manuscript.
- Introduction: Line 33-35: "In a single sample, a given surface parameter across a wide range can be systematically explored, avoiding the need for lengthy repetitive procedures and enhancing the efficiency of research and development." This sentence was not clear to me and requires citation(s).
- Line 57 - 59: "However, the GPC method ...on flat substrate". Please add references for the claim.
- Line 61: fix the typo “polymes”
- Line 112 - 114: "During relaxation, a randomly ... avoiding bond intersections". Please rewrite the sentence to bring out the meaning.
- Line 120 -122: To keep the paper self-contained, and since it is one of the major methods simulated in this paper. The authors should try to at least explain succinctly the difference and then cite the reference for further details.
- Please add axis labels in scheme 1a (x & z) and b (x & y).
- Line 124 - 126: Why "the initiators exhibited a random distribution" is not clear to me. The grafting density is sigma + [delta sigma], which equals a constant value in a given stripe. Thus, the number of points at y = 1 in the stripe of width w is also a constant. To my understanding, the authors wanted to explain that the placement of the initiators is random. But in a given stripe, based on the grafting density (sigma) value of that stripe, the total initiator points are constant, and not random. If that is the case, please rewrite the sentence.
- Line 124: of size w*Lz, as illustrated in Scheme 1. w = 4 was stated in line …, w*Lz is not equal to the dimension stated. Please clarify.
- Line 127 - 129: "Due to the periodical boundary condition in x direction, the grafting density mirrored from the left to the right part of the simulation box, i.e., the grafting density decreased from the middle to the rightmost stripe as x increased further." Maybe I am missing something, I do not understand the rationale for such a design choice. The authors claim in line 197 that the steepness gradient is [delta \sigma]/w where w = 4 but for the central region (sigma_max) and the extreme ends (left and right) \sigma_max w becomes 8. Why the authors did not choose a simulation box that was symmetric where the left and right ends could have been of w/2 size or they could have just ignored one of the ends preferably the last stripe at the right end at sigma_min + d\sigma (periodicity would have accounted for the other end), and keeping the central sigma_max of the same width as w instead of 2*w as shown. This would have correctly captured the periodic symmetry with a uniform width of w for each strip, and the steepness gradient value would have been a constant of [delta \sigma]/w where w = 4.
- In the results section, the authors make use of the headings in 3.1 and 3.2 to distinguish the different simulations carried out. I suggest the authors rewrite lines 93 - 98 so that it provide easy referencing to readers.
- Line 161: Please re-write the sentence to bring out the meaning. “The decrease is more obvious as the polymerization processes. ”
- Typo Line 162 sigma = 0.1 instead of 0.01
- The paragraph (lines 78 -92) mentions the previous research and the difference. However, the authors carry out section 3.1 to reinforce the idea.
- To me, the results in sec 3.1 seem obvious. Maybe I am missing something, but to me, in the homogeneously distributed scenario, increasing sigma will produce lower molecular weight polymers. Since the monomer pool was kept constant at [M]_0 = 0.4? Further, their lattice model has excluded volume. It is not clear to me whether the model considers entanglement. Further, the dispersity too will increase if the initiator density increases. Thus, Figure 1 seems to be more a validation of their modeling framework than new insights.
- Please plot the final time for MWD distribution. Also, the plot for \sigma = 0.1 does not seem to saturate by the simulation timesteps since the curve seems to be still rising (Fig 1a).
- Why the error bars are not plotted in Figures 1, 3, and 4?
- Lines 167 - 175 should be explained more clearly. The claim made in eq. 3 is noticed only for a very short time in Fig. 1a t ~ 10^5 after that the plots diverge. Thus, the explanation provided by the authors is not clear to me.
- Line 199 - 200: The polymerization stops when the number-average molecular weight of the brush reaches 50. Why? Is this based on some assumption/observation?
- In sec 3.2, what was the monomer concentration? How were the monomer sites chosen in each of the stripes? If every stripe has [M]_0 = 0.4 then the strip at the middle has higher grafting density = 0.42 compared to the monomer fraction. It is advised that the authors provide additional details in terms of numbers maybe in the SI. Further, why the monomer fraction was not varied?
- If the monomer fraction is close to 1. Do the authors still observe the low value of scaling ~ 0.15?
- Please state the assumptions and limitations of the modeling approach. For example, am I correct to interpret: In the model, the monomers are initially placed randomly based on monomer concentration at different positions? These monomers do not change places in the entire duration of the simulation. Diffusion is not accounted for in the simulation. The vacant sites stay vacant for the entire duration of the simulation. Thus, if the neighbor bonds are created and since bond-intersection and excluded volume are accounted for, what is the probability that some of the chains at higher grafting density initially grow and then cannot grow further because of low monomer availability and bond criteria hindering the growth of these chains?
- Fig2. The simulated system is 3D but the plots shown are 2D. Please describe clearly how the plots were made. If possible add a 3D density plot.
- Please plot Fig.3 and 4 with other values of monomer concentration to show the effect of monomer concentration preferably slightly higher and very high compared to initiator density.
- Caption Fig. 4a check spelling "cycle" should be "circle" or "disks".
- I did not find any explanation for Fig. 4b.
- The SI is only one page long. The authors can explain their SRM model in the SI to make the paper self-contained.
Comments on the Quality of English Language
English is alright. Some sentences need to be re-written to bring out the meaning.
Reviewer 2 Report
Comments and Suggestions for Authors
The authors study gradient brushes with lattice Monte Carlo simulations. They observe a novel scaling regime that is not predicted by theory nor observed experimentally. They claim that polydispersity and spatial heterogeniety cause a reduction in scaling exponent. It is argued that experiments by Genzer that do reproduce theoretical scaling exponent is "coincidental" since polydispersity was ignored.
It would appear from Fig.4d that the theoretical exponent (α = 1/3) is obtained for monodisperse brushes in the limit of zero initiator gradient. However this result seems not discussed anywhere. Could polydispersity be introduced systemically, to show that this lowers the scaling exponent? In particular, it would allow an assessment of the reduction in exponent for the polydispersity in the experiments by Genzer (1.7).
Fig. 4d indicates that the scaling exponent depends strongly on the initiator gradient for monodisperse polymer. Has a similar variation in applied gradient be made for the gradient brush?
Does the observed exponent show a system size dependency?
How does the density profile compare with theoretical predictions for polydisperse brushes, e.g. those in Milner et al (ref 48).
In Figure 1c, the caption (line 188-189) mentions "dotted lines". Are they missing or do they coincide with the simulation data?
Line 53: typo in interrelatted
Line 223: it seems the symbol σ is missing in "as a function of in both"
Line 234: sentence appears in bold
Round 2
Reviewer 1 Report
Comments and Suggestions for Authors
The authors have responded to the queries properly, and I recommend the manuscript for publication. However, I would like the authors to include some of the responses from the response document in the main manuscript which are detailed below.
Response 17 - 22: Please include the details from the response document to the main manuscript. This will help make the manuscript clear, detailed, and easier to understand, and help in reproducible research for the research community.
Comments on the Quality of English Language
Please proofread carefully the main and SI of the manuscript. Some sentences have small grammatical errors such as the Fig. 3 caption in the SI.
Author Response
Thank you very much for confirming our revised manuscript and providing help suggestions. Following your suggestions, we have included some of the responses to the main text in the latest version, such as the concerns on entanglement, Mn=50, saturation of Mn with grafting density 0.1, and the illustration of the SRM. These changes are indicated by the track-change mode.
Additionally, we have tried our best to correct typos and grammatical errors.
Best regards,
Shichen